# Hydrology of inland tropical lowlands: The Kapuas and Mahakam wetlands

H. Hidayat[1,2], A.J. Teuling[1], B. Vermeulen[1], M. Taufik[1,7], K. Kastner[1], T.J. Geertsema[1], D.C.C. Bol[1], D.H. Hoekman[3], G.S. Haryani[2], H.A.J. Van Lanen[1], R.M. Delinom[4], R. Dijksma[1], G.Z. Anshari[5], N.S. Ningsih[6], R. Uijlenhoet[1], and A.J.F. Hoitink[1]

[1]Hydrology and Quantitative Water Management Group, Wageningen University, Wageningen, The Netherlands
[2]Research Center for Limnology, Indonesian Institute of Sciences, Cibinong, Indonesia
[3]Earth System Science and Climate Change Group, Wageningen University, The Netherlands
[4]Research Center for Geotechnology, Indonesian Institute of Sciences, Bandung, Indonesia
[5]Soil Science Department, Tanjungpura University, Pontianak, Indonesia
[6]Faculty of Earth Sciences and Technology, Bandung Institute of Technology, Bandung, Indonesia
[7]Department of Geophysics and Meteorology, Bogor Agricultural University, Bogor, Indonesia

*Correspondence to:* H. Hidayat (hidayat@limnologi.lipi.go.id)

**Abstract.** Wetlands are important reservoirs of water, carbon and biodiversity. They are typical landscapes of lowland regions that have high potential for water retention. However, the hydrology of these wetlands in tropical regions is often studied in isolation from the processes taking place at the catchment scale. Our main objective is to study the hydrological dynamics of one of the largest tropical rainforest regions on an island using a combination of satellite remote sensing and novel obser-
vations from dedicated field campaigns. This contribution offers a comprehensive analysis of the hydrological dynamics of two neighbouring poorly gauged tropical basins; the Kapuas basin (98,700 km$^2$) in West Kalimantan and the Mahakam basin (77,100 km$^2$) in East Kalimantan, Indonesia. Both basins are characterized by vast areas of inland lowlands. Hereby, we put specific emphasis on key hydrological variables and indicators such as discharge and flood extent. The hydro-climatological data described herein were obtained during fieldwork campaigns carried out in the Kapuas over the period 2013-2015 and in
the Mahakam over the period 2008-2010. Additionally, we used the Tropical Rainfall Measuring Mission (TRMM) rainfall estimates over the period 1998-2015 for analysing the distribution of rainfall and the influence of El-Niño – Southern Oscillation. Flood occurrence maps were obtained from the analysis of the Phase Array L-band Synthetic Aperture Radar (PALSAR) images of 2007 through 2010. Drought events were derived from time-series of simulated groundwater recharge using time series of TRMM rainfall estimates, potential evapotranspiration estimates and the threshold level approach. The Kapuas and
the Mahakam lakes region are vast reservoirs of water of about 1000 km$^2$ and 1500 km$^2$ that can store as much as 3 billion m$^3$ and 6.5 billion m$^3$ of water, respectively. These storage capacity values can be doubled considering the area of flooding under vegetation cover. Discharge time-series show that backwater effects are highly influential in the wetland regions, which can be partly explained by inundation dynamics shown by flood occurrence maps obtained from PALSAR images. In contrast to their nature as a wetland, both lowland areas have frequent periods with low soil moisture conditions and low groundwater recharge.
The Mahakam wetland area regularly exhibits low groundwater recharge, which may lead to prolonged drought events that

can last up to 13 months. It appears that the Mahakam lowland is more vulnerable to hydrological drought leading to a more frequent fire occurrence than the Kapuas basin.

## 1 Introduction

Lowland rivers are of major importance for mankind not only due to their extensive uses for many purposes such as food production, drinking water, and navigation, but also they are important ecosystems that thrive from regular supply of nutrients from sediment deposited during floods. Since the very beginning of agricultural activities, most of the worlds' population is concentrated on fertile alluvial floodplains that support food production as well as access to waterways for transport (Verhoeven and Setter, 2010). Envisioning the fate of lowland rivers and the adjacent wetlands in response to current threats including those stemming from natural processes as well as of anthropogenic origin will likely remain a challenge for the coming years. Measures to mitigate flood, drought, and loss of biodiversity add to elements that keep lowland rivers a dynamic theme (Hoitink and Jay, 2016).

Typically, lowlands are located in the downstream part of a river basin in the form of deltas. However there are also lowlands that are located in more upstream parts of the basin. The absence of topographical gradients often leads to the formation of (seasonal) wetlands. The Congo and Amazon are the best known examples of tropical rainforest regions characterized by vast areas of lowlands and wetlands (Hess et al., 2015). Beside permanent open water lakes, the Congo basin contains extensive wetland system with diffuse water shorelines and in some areas inundation are under thick vegetation cover, which form one of the largest swamp forest in the world known as the Cuvette Centrale (Alsdorf et al., 2016). However, there is a lack of knowledge on processes involved in seasonal flooding in tropical lowland areas. Inundation extent, water depth, groundwater, as well as storage volume of wetland are not well known as few hydrological studies have been conducted in this area (Alsdorf et al., 2016). The floodplain of the Amazon River also contains lakes that are temporally or permanently connected to the main river by several channels (Bonnet et al., 2008). Water from the Amazon flows into the floodplain lakes at the beginning of the rising water, but by mid-rising water, lake water gradually flows out into the river (Lesack and Melack, 1995). Along with these processes, groundwater regulates the seasonal dynamics of the Amazon surface waters (Miguez-Macho and Fan, 2012). Hoch et al. (2017) recently asses the impact of hydrodynamics on flood wave propagation of the Amazon by coupling a global hydrologic model with a hydrodynamic model. They found that although the coupled runs simulate discharge better than hydrology-only runs, some peak flow are overestimated due to the lack of a feedback loop to hydrological processes on floodplains, such as evaporation and groundwater infiltration.

Although definitions of wetlands are based on hydrologic conditions (Zedler and Kercher, 2005; Verhoeven and Setter, 2010), hydrology is key aspect of wetlands that is the most poorly described due to technical, cost, and time constraints of field measurements in this area (Cole et al., 1997). Determining an average or specific hydroperiod, the period when a soil area is waterlogged, or the opposite, a period with below normal water storage for wetland sites, subsequently remains a major technical challenge. Piezometers are commonly used to analyze water levels in studies at the plot scale in peatlands, however, maintaining them are labour intensive (Devito et al., 1996; Fraser et al., 2001; Baird et al., 2004; Ishii et al., 2015). The

application of satellite remote sensing techniques is one approach to address this scarcity of information. Data from optical sensors such as Landsat imagery can be used for this purpose in areas with little cloud cover (e.g. Qi et al., 2009; Adam et al., 2010; Ran and Lu, 2012). However, it is hard to satisfy the preferred limit of cloud cover for such images covering the humid tropics. For this reason, radar remote sensing is considered as the most suitable technique in land and water observation in

tropical regions. Hidayat et al. (2012) demonstrate that flood occurrence information and its corresponding extent of open water as well as areas under vegetation cover can be extracted from a series of images from the Phase Array L-band Synthetic Aperture Radar (PALSAR) covering a humid tropics lowland site. Using images from an L-band SAR sensor, Hess et al. (2015) exhibit the capabilities radar remote sensing technique for mapping tropical wetland extent and inundation over large regions. Other approaches that try to overcome at-site data scarcity involve simulation modeling using large-scale datasets as input (e.g.,

Van Loon et al., 2012).

Lowland regions are typified by certain hydrological properties, e.g. small hydraulic gradients, shallow groundwater, a flat topography, and a high potential for water retaining in wetlands (Schmalz et al., 2008). Intensive groundwater-stream water interactions as well as soil moisture-groundwater interactions occur in these areas as a result of shallow groundwater conditions (Brauer et al., 2014). The hydrology of lowlands is complicated by, among others, backwater effects, lake-river interaction,

possible tidal effects and hydrological extremes. Backwater effects result in an ambiguous stage-discharge relation (Petersen-Overleir and Reitan, 2009; Herschy, 2009; Hidayat et al., 2011b), as such that at any given discharge, falling river stages being much higher than rising stages. Tides have a significant impact on the river flow farther away from the river mouth in lowland regions by means of subtidal water level variations controlled by river-tide interactions (Buschman et al., 2009; Hoitink and Jay, 2016). Therefore, hydrological tools such as the use of rating curves, rainfall-runoff models and flood prediction may fall

short if they are applied without a proper adaptation. Novel measurement and modelling methods such as continuous flow measurements (Sassi et al., 2011; Hoitink et al., 2009) and neural networks (Corzo et al., 2009; Hidayat et al., 2014) can be a promising means concerning the complex interactions of peat areas, lakes, runoff, and tides in this region. Although tropical wetlands are more prone to high water, these also regularly suffer from drought (Walsh, 1996). This requires approaches that consider both hydrological extremes.

The Kapuas and Mahakam river basins on Kalimantan represent two large tropical lowland areas. Fig. 1 shows the location of the Kapuas (total catchment area of 98,700 km$^2$) in West Kalimantan, and the Mahakam (77,100 km$^2$) in East Kalimantan. Both rivers are among the longest rivers worldwide found on an island and are characterized by vast areas of inland lowlands. The size of the rivers, the complex geomorphology of their lowland channel networks, and the hydrological links with the adjacent peat bogs and inland wetlands, which are prone to drought and forest fires during dry years, render the Kapuas and

the Mahakam basins a scientific challenge to study. The Kapuas and Mahakam Rivers exemplify data poor environment in the tropical region, with catchments that comprise vast areas of rain forest. Few hydrological studies have been conducted in the region despite the rivers' importance to environment and the people. Part of these catchments can be considered relatively pristine, offering a view on a natural hydrological regime which serves both scientific and engineering purposes. Compared to their mid- and high-latitude counterparts, few studies have addressed the hydrological dynamics of large tropical rivers.

This is mainly caused by the fact that most tropical rivers are poorly gauged. Over the past decade, many hydrological studies

have focussed on the problem of ungauged basins under the 'Prediction in Ungauged Basins' initiative (Sivapalan et al., 2003; McGlynn et al., 2013) of the International Association of Hydrological Sciences'. Through growing international research connections that attempt to holistically study the terrestrial system and the development of globally consistent databases, including those from remote sensing observations, climate stations, downscaled bias-corrected output from climate models, this gap is now gradually changing (Syvitski et al., 2014; Harris et al., 2014; Weedon et al., 2011). This contribution contributes to this trend. By combining field measurements of key hydrological variables over two large tropical catchments with modeling and different sources of satellite remote sensing, we can quantify flooding both in terms of water level, inundated area and volume, and we reveal the impacts important processes such as backwater effects.

Our central objective is to study the hydrological dynamics in the Kapuas and Mahakam river basins, located in of one of the largest tropical rainforest regions outside the Congo and Amazon using a combination of satellite remote sensing and novel observations from dedicated field campaigns. Hereby, we put specific emphasis on key hydrological variables and indicators such as discharge and flood extent. Resolving hydrological processes is essential to understand the impact of changes in terrestrial hydrological and biogeochemical cycles including land degradation on water level dynamics, water quality, and ecology of these important yet vulnerable wetland regions. The interactions between wetlands and the river have implications for geomorphology, governing sediment retention and modulating peak discharges, and for estuarine processes, controlling salinity intrusion during low flow. The dataset will offer a solid database which will find its use in future research and engineering. Section 2 of this paper describes the two study catchments. Section 3 presents field data gathering to measure water levels, soil moisture, and discharge. This Section also describes satellite-based data that cover the entire catchments and the method used for obtaining inundation maps. Section 4 presents the results in the form of a hydrological comparison between the two lowlands. Section 5 presents the discussion and conclusions are drawn in Section 6.

## 2 Study area

The Kapuas and the Mahakam are the longest and the second longest rivers in Indonesia with the length of about 1,140 and 980 km, respectively.The Kapuas and the Mahakam lowlands are located about 650 and 250 km from their respective river mouths. Lake Sentarum National Park in the upper Kapuas River is an important Ramsar site, which represents one of old tropical peat formations in Late Pleistocene (Anshari et al., 2001, 2004). The Mahakam River is home to endemic species including the Irrawady dolphin (*Orcaella brevirostris*), which is listed as 'Vulnerable' on the International Union for Conservation of Nature (IUCN) Red List of Threatened Species due to, among others, entanglement in gillnets, vessel traffic, sedimentation, habitat loss and degradation from habitat change (IUCN, 2016). The Kapuas and the Mahakam wetlands are important for their respective local communities not only as a source of water for domestic purposes, but also to sustain the livelihood of the people especially in the open water fishery sub-sector. The Kapuas wetland with its seasonally inundated lakes produces about 18,000 tons of freshwater fish annually (BPS-Kalbar, 2015). The Middle Mahakam wetland is the core of inland fisheries in East Kalimantan and is considered as one of the most productive freshwater fisheries in Southeast Asia (MacKinnon et al., 1996) with a current estimated annual fish production of 33,000 tons (BPS-Kaltim, 2015). These fishing industry figures express the

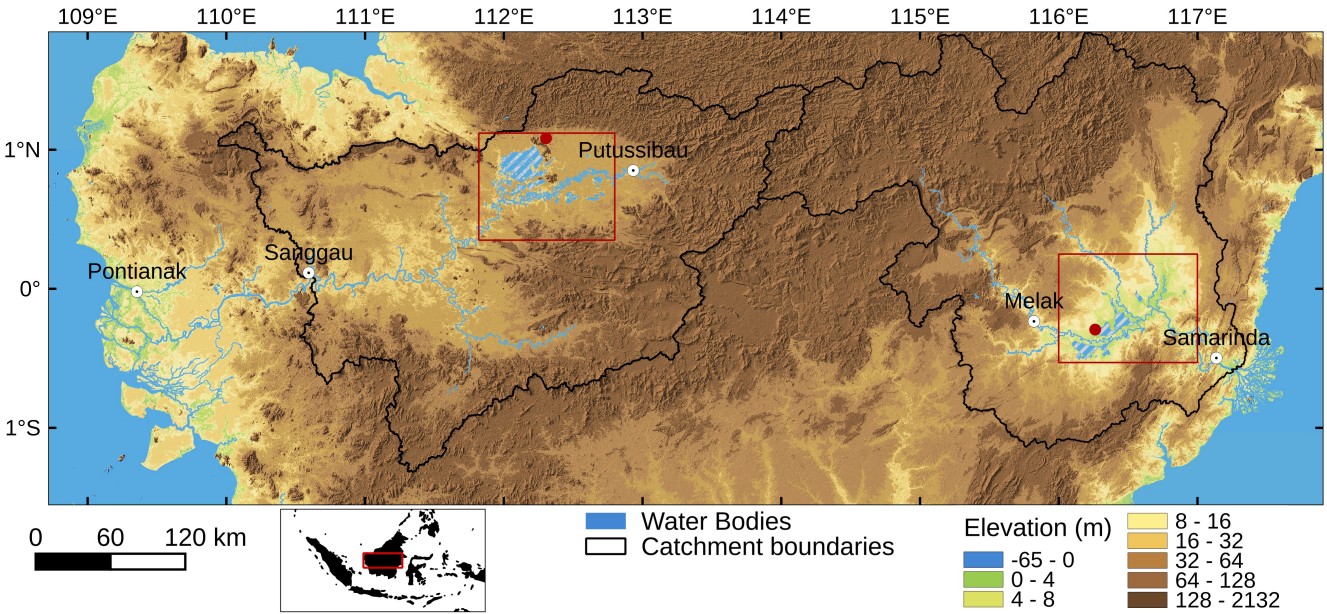

**Figure 1.** Map of the research area. Outlined are the Kapuas catchment upstream of Sanggau (left) and the Mahakam catchment upstream of its delta region (right), indicated by the black line. Red boxes indicate the focus regions encompassing the Kapuas and Mahakam lowlands. Red circles indicate the locations of point rainfall, soil moisture, and groundwater observations. Discharge monitoring stations are located in the city of Sanggau (the Kapuas) and in the city of Melak (the Mahakam).

high economic value of the wetland areas we study. The unique tropical wetland ecosystems are rich in biodiversity of typical aquatic as well as terrestrial flora and fauna, which is why they are listed as a Ramsar site, urging conservation. Notwithstanding their ecological, hydrological and economical importance, the Kapuas and Mahakam wetlands in particular and the two river basins in general have been increasingly threatened by a variety of factors including pollution, forest fires, deforestation, and
5  mono cultures (Rautner et al., 2005; Chokkalingam et al., 2005).

Both the Kapuas and Mahakam rivers originate from the centre of Borneo on the border between West Kalimantan (Indonesia) and Sarawak (Malaysia). The Kapuas originates from the Kapuas Hulu Mountains runs westwards through mountainous terrain and descending into a flat plain (Loh et al., 2012). At this plain, from Putussibau until the delta, the river elevation drops by just 50 m over a length of 900 km (MacKinnon et al., 1996). This results in the formation of the Kapuas floodplain lakes
10 area in the upstream part of the river. Figure 2 (top pictures) show conditions of the lakes area during wet and dry periods. The Kapuas lake area that includes the Lake Sentarum National Park is surrounded by mountain ranges; the Upper Kapuas Mountains in the North, the Muller Mountains in the East, the Madi plateau in the South and the Kelingkong mountains in the West (MacKinnon et al., 1996). Despite its inland position, most of this wetland basin lies less than 30 m above sea level (Anshari et al., 2004). Downstream of the wetlands, the Kapuas meets a main tributary, the River Melawi, flows further westwards
15 through a low mountain land, bifurcates into the Kapuas Besar and the Kapuas Kecil, and finally continues to the Kapuas delta

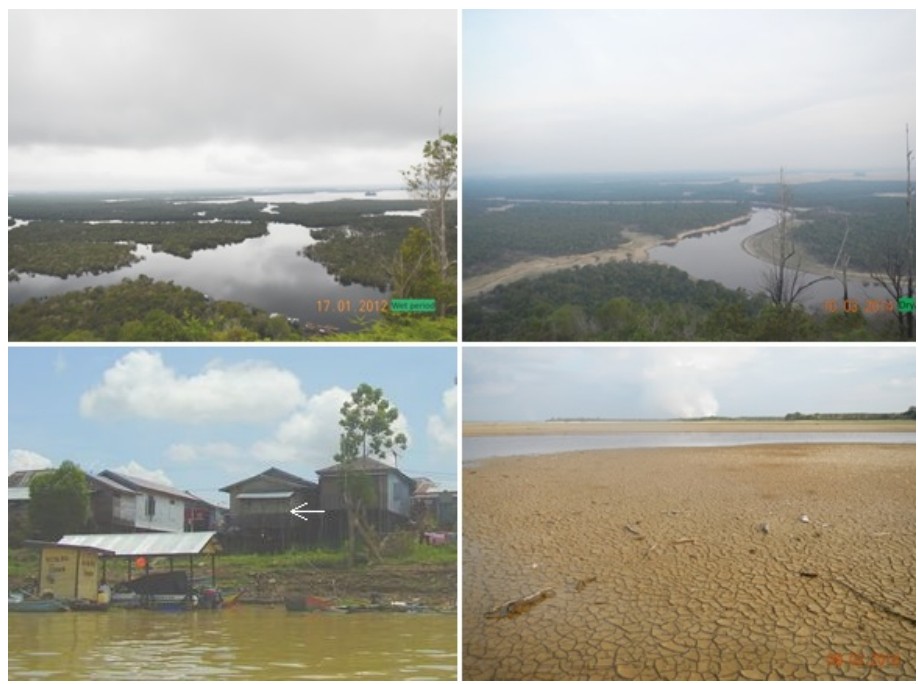

**Figure 2.** Photos from a wetland reconnaissance: (top left) The Kapuas wetlands during wet period showing full inundation and (top right) during dry period showing the exposed lake bed and the remaining wetted lake channel. (bottom left) The Mahakam at the lakes area; flood marks on trees and houses (white arrow) show the highest water level. (bottom right) Another scene from a drought event in the Kapuas wetlands with forest/shrub fires on the background.

distributaries towards the Karimata Strait. This strait is connected to the South China Sea, which is very busy and important for livelihoods of communities in Southeast Asia. The capital of West Kalimantan, Pontianak, which is inhabited by about 600,000 people (estimate from 2014, see BPS-Kalbar, 2015), lies right on the equator along the Kapuas Kecil.

The Mahakam flows from its source through the Pre-Tertiary rocks in south-eastwards direction, reaches the Tertiary basin of Kutai. The river then meets the Kedang Pahu River in the middle Mahakam region (Van Bemmelen, 1949). The Mahakam meanders from there eastward through Quarternary alluvium in the Mahakam lakes area. This area is a flat tropical lowland region circumscribed by wetlands with seasonal flooding (Fig. 2, bottom left picture). Some thirty shallow lakes, which are connected to the Mahakam through small channels, are situated in this area. The Mahakam meets three other main tributaries downstream of the lakes area and flows south-eastward through a Tertiary mountain range before reaching the Mahakam delta distributaries and debouches into the Makassar Strait. Samarinda, the capital of East Kalimantan, which is inhabited by over 830,000 people (estimate from 2014, see BPS-Samarinda, 2015), lies along the Mahakam just upstream of the delta region.

According to Köppen climate classification the climate of the region is tropical rain forest (Af type), which is characterized by a long-term mean precipitation higher than 60 mm in the driest month. The regional climate of Kalimantan is generally influenced by the Indo-Australian monsoon driven by the Intertropical Convergence Zone (ITCZ) and El-Niño – Southern

Oscillation (ENSO) phenomena (Meehl and Arblaster, 1998; Seidel et al., 2008). The central and northern parts of Kalimantan have bimodal rainfall patterns with two peaks of rainfall (generally occur in October through November and March through April) due to the global circulation and the regional climate (Aldrian and Susanto, 2003). The development of El-Niño in the Pacific indicated by an anomalously cold Sea Surface Temperature (SST) surrounds Indonesia while warm anomalies

develop in the eastern Pacific and western Indian Ocean (Hendon, 2003) generally triggers drought conditions in the region. Conversely, the development of a La Niña event indicated by SST anomalies opposite to those during an El-Niño event results in increased rainfall during the dry season. However, the ENSO effect in Indonesia is not uniform throughout the seasons. Rainfall anomalies tend to persist in the dry season but not in the wet season (Hendon, 2003).

## 3   Methodology

Field data described in this contribution were obtained during fieldwork campaigns in the Mahakam held in 2008—2009 and in the Kapuas held in 2013—2015. Details on field data used in this study are presented in Table 1. Point measurements of rainfall were conducted using rain-gauge and automatic weather station. The Tropical Rainfall Measuring Mission (TRMM) rainfall estimates were acquired from the internet site of NASA's Goddard Earth Sciences Data and Information Services Center Interactive Online Visualization and Analysis Infrastructure (Giovanni) at http://giovanni.gsfc.nasa.gov/. We used the

TRMM daily 3B42 and monthly 3B43 rainfall products in this study. The daily TRMM rainfall rate was used to show the spatial rainfall variability in the study area and to be compared with previous studies (Wuis, 2014; Tekelenburg, 2014) with short period of observation (about two months). The monthly TRMM rainfall rate was used to be presented along with the publicly available SOI index, which has monthly values. A comprehensive description of the TRMM rainfall products, which have been available since 1998, can be found in Huffman et al. (2007). Water levels were measured using pressure transducers,

while soil moisture sensors were used to estimate the volumetric soil moisture contents. Water levels of lakes were measured at the shore, resulting in an unmeasured water column of the deeper region. Therefore, a depth of 1 m was added to the water levels for water volume estimation, based on bathymetry measurements. Discharges were estimated from measurements using the Horizontal Acoustic Doppler Current Profilers (H-ADCP) deployed in the middle Mahakam at Melak, which is upstream of the Mahakam wetland region about 300 km from the river mouth (Hidayat et al., 2011b) and in the middle Kapuas at Sanggau,

which is situated about 270 km from the river mouth (Kastner et al., 2015). The discharge data from H-ADCP measurements were in half-hourly time step that enable us to see the hysteretic behaviour of discharge as a result of backwater effects. To establish water discharge through the river section, boat surveys were carried out at the cross-sections where the H-ADCP was deployed.

The Sanggau flow monitoring station is located downstream of the Kapuas wetland region. Therefore, to evaluate the water

surface profile and flow from this discharge station upstream through the Kapuas wetlands region, a one-dimensional hydraulic representation of the river was made in HEC-RAS (US Army Corps of Engineers, 2002), an open software river analysis system developed by the US Hydrological Engineering Centre. Two of the HEC-RAS components were used; the steady flow surface water computations (to determine average water profiles and levels for minimum and maximum discharges) and the

**Table 1.** Description of collected field data.

| Measurement station | Location | Period of data | Variable measured |
|---|---|---|---|
| **Mahakam** | | | |
| Rain-gauge at Melintang | 116.2563 E; 0.3016 S | May2008–Mar2009 | rainfall, air temperature |
| Pressure transduser at Melintang | 116.2585 E; 0.2932 S | Feb2008–May2009 | groundwater level |
| Pressure transduser at Jempang | 116.1884 E; 0.4959 S | Mar2008–May2009 | lake level |
| H-ADCP Discharge station at Melak | 115.8684 E; 0.2998 S | Mar2008–Aug2009 | water flow velocity, water level, local bathymetry |
| **Kapuas** | | | |
| Automatic weather station at Leboyan | 112.2909 E; 1.0830 N | Dec2013–Mar2015 | rainfall, air temperature |
| Pressure transduser at Leboyan | 112.3094 E; 1.0876 N | Dec2013–Mar2015 | groundwater level |
| Pressure transduser at Sentarum | 112.0636 E; 0.8388 N | Mar2014–Mar2015 | lake level |
| Soil moisture sensor at Leboyan | 112.2909 E; 1.0830 N | Dec2013–Feb2014 | soil moisture content |
| H-ADCP Discharge station at Sanggau | 110.5889 E; 0.1149 N | Oct2013–Apr2015 | water flow velocity, water level, local bathymetry |

unsteady flow simulation (to simulate one-dimensional unsteady flow through a full network of open channels). Upstream of the Kapuas and the Melawi River (the southern tributary) a flow hydrograph was given as boundary condition, based on characteristics of the upstream sub-catchment that drains directly into the Kapuas River. At the downstream end of the river, a rating curve function obtained from the stage-discharge relation at Sanggau was provided as boundary condition. We used a constant Manning roughness coefficient $n$ of 0.035, which corresponds to a normal river channel with some weeds and stones for the entire Kapuas and Melawi reaches. A spatial discretization of 3 km and a time step of 20 minutes were chosen for this model. Spatial resolution of 3 km is considered sufficient for the modelled river section of 553 km from Putussibau to Sanggau. Regarding time resolution, flow velocity and cross-sectional distance show that the resolution in time cannot exceed 25 minutes. Models runs showed that the model is stable for $\delta t < 20$ min. For $\delta t$ equal to 30 minutes, the model was conditionally stable, and it is unstable for $\delta t > 1$ hr. The difference between model outcomes on a 20 minute resolution vs a 1 minute resolution was negligible. Model evaluation was carried out using the Nash-Sutcliffe (NS) efficiency coefficient:

$$NSE = 1 - \frac{\Sigma_{t=1}^{T}[Q_{obs}(t) - Q_{sim}(t)]^2}{\Sigma_{t=1}^{T}[Q_{obs}(t) - \bar{Q}_{obs}]^2} \tag{1}$$

where $Q_{obs}(t)$ and $Q_{sim}(t)$ correspond to observed and calculated discharge at time $t$, $\bar{Q}_{obs}$ is the average observed discharge, and $T$ is the total number of time steps. NS coefficient for water level simulation was calculated in the same way by changing $Q$ with $h$.

Our HEC-RAS model was not entirely calibrated; literature values were used for parameters which were not measured. Uncertainty in model parameters can have an effect on the model outcomes. Therefore, effects of changes in lateral influx, lake water storage and Manning's roughness coefficient ($n$) were investigated (Table {tab:sim). Water stage and discharge at Sanggau station and lake water level data were used to validate HEC-RAS model simulations. An increase in lateral fluxes

**Table 2.** Effects of changing lateral influx ($Ql$) [$m^3s^{-1}$], lake storage/elevation ($L$) [m] and Manning's $n$ [$sm^{-1/3}$] on simulated water level ($h$) and dicshrge ($Q$) at Sanggau.

| Scenario | NS $Q$ Sanggau | NS $h$ Sanggau |
|---|---|---|
| $Ql$ 1.1 | 0.81 | 0.09 |
| $Ql$ 0.9 | 0.87 | 0.77 |
| $L$ 1.1 | 0.88 | 0.59 |
| $L$ 0.9 | 0.88 | 0.50 |
| $n$ 0.03 | 0.89 | 0.48 |
| $n$ 0.03 | 0.87 | 0.53 |

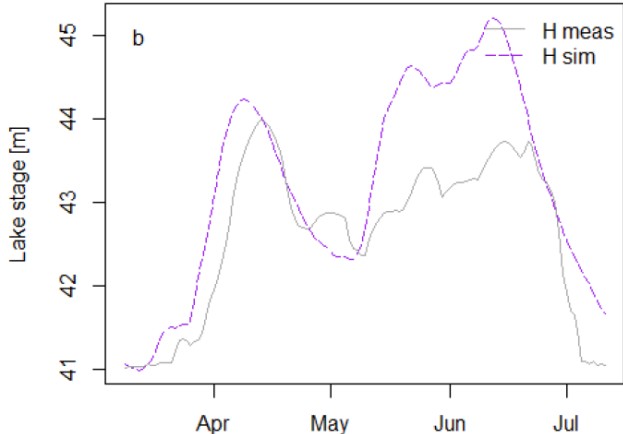

**Figure 3.** Simulated (dashed line) and measured (solid line) water levels (m above sea level) of the Kapuas lake.

leads to an increase in peak flow, lake level fluctuation and river stage fluctuation. An increase in storage vs elevation relation, decreases the lake water depths, but increases the outflow as the lake contains more water with the same elevation. A decrease in $n$, a decrease in bottom friction, leads to faster and larger discharges; the discharge at Sanggau increases. The associated river stages over the entire river profile are lower, as water discharges faster. Wetland water stages are also lower when the

5  Manning's n decreases. Table 2 also shows that an increase in NS of the discharge simulation at Sanggau is associated with a decrease in NS of the stages simulation at Sanggau for all scenarios. Figure 3 shows the simulated and measured water levels of the Kapuas lake. The Kapuas wetland was modelled as one large reservoir. In fact, it is a complex system of seasonally connected small lakes and peatland that may cause discrepancies in magnitude and timing of changes in water levels of the lake. Our model simulation reflects the bimodality of the rainfall pattern in the study area, as shown in the simulated discharge

10  (Fig. 4).

Inundation maps were obtained from the analysis of PALSAR images available for the period of 2007—2010. Flood occurrences were mapped following the method for flood mapping of open water and flooding under vegetation (Hidayat et al.,

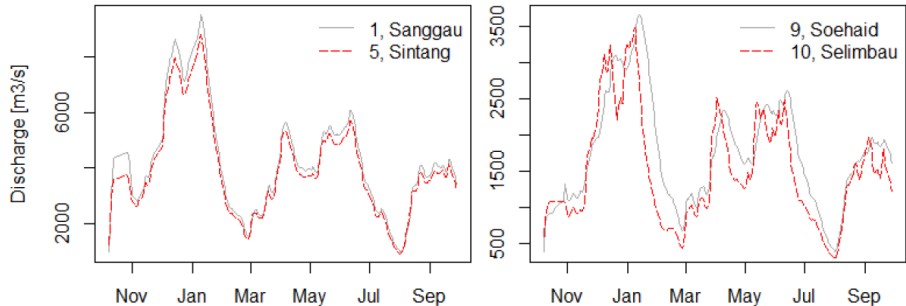

**Figure 4.** HEC-RAS simulated discharges for different cross sections in the downstream Kapuas River (left) and the upstream Kapuas River (right) of the modelled river section.

2012). Crucial in this method is the onset of backscatter for open water (upper threshold) and for inundation under vegetation cover (the lower threshold). To determine threshold values for open water flood occurrence mapping, radar backscatter statistics of regions covering the main river and lakes known to be permanently inundated were taken. The mean plus one standard deviation was chosen as the upper threshold for inundation as open water. For inundation under vegetation, the mean value of radar backscatter sampled from floodplain regions known to be frequently flooded was taken as the lower threshold. Open water inundation occurrence has an accuracy of about 85% while the reported accuracy of inundation occurrence under vegetation is 78% (Hidayat et al., 2012). From inundation area and lake water levels, we roughly estimated the total volume of water in the lakes region. Areas of floodplain lakes are following the seasonal inundation pattern as a result of variable rainfall rates. The period of Mahakam water level measurements coincided with part of the PALSAR data acquisition dates. Therefore, we were able to derive a lake depth ($h$) - area ($A$) relationship that was used to estimate the lakes' water levels beyond the period of our measurements (Fig. 5). For the Kapuas, our water level measurement did not coincide with the PALSAR observation period. With an assumption of uniform distribution of depths, we develop a depth-rainfall relationship. Correlation analysis reveals that the depth of the lakes is well-correlated with the two-month moving average of areal rainfall ($P_{2m}$). We obtained a linear depth–rainfall relation that was applied to approximate water levels of the Kapuas lakes, reading:

$$h = 0.021 * P_{2m} - 2.4. \tag{2}$$

For a drought study in the two basins, we simulated the transient water balance (Van Lanen et al., 2013) using rainfall estimates from TRMM and potential evapotranspiration from the Climate Research Unit (CRU) (Harris et al., 2014) for the period from 1998 through 2014 as input data to derive groundwater recharge. The spatial resolution of the gridded potential evapotranspiration obtained from CRU is 0.5 degree. Our measured data were used to validate the TRMM product. Therefore, the difference in the period of field campaigns in the Kapuas and Mahakam basins has little effect on the overall results. Different land uses to simulate the actual evapotranspiration in the wetland area were identified from the Borneo land-use and land cover map (Hoekman et al., 2010). Drought events are derived from time-series of groundwater recharge using the threshold level approach. This approach defines drought as a period when the recharge is below a certain threshold value (Hisdal

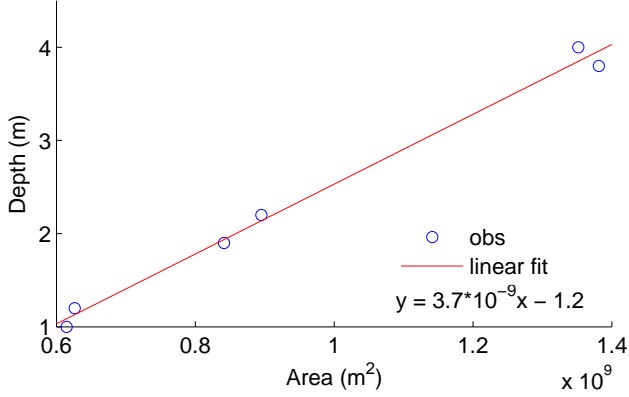

**Figure 5.** Relationship between measured depth of the middle Mahakam lakes and area of open water inundation from PALSAR images.

et al., 2004). We applied, as done in previous studies (e.g. Sheffield and Wood, 2008; Tallaksen et al., 2009; Van Lanen et al., 2013), the 80 percentile from the duration curve as a threshold. We applied monthly variable thresholds that were identical for both river basins, which were derived from the Mahakam basin being the driest. Drought duration is considered as the period from the start to the end of an uninterrupted drought.

## 4   Hydrological characterization

### 4.1   Climate and climate variability

Rainfall rates are generally higher in the Kapuas than in the Mahakam. The spatial distribution of rainfall as TRMM daily rainfall rate (mm) averaged over January 1998 – December 2014 is shown in Fig. 6. The lowest average rainfall rate of about 5 mm/day was found in the Mahakam wetland region, while the highest rainfall rate of about 11 mm/day was found in the upper Kapuas region. As the spatial resolution of the TRMM data is very rough, small scale variation cannot be observed. Spatial variability of rainfall is obvious in a sub-catchment scale daily field observation. Wuis (2014) and Tekelenburg (2014) reported that most high intensity showers in Bika, a sub-catchment in the upper Kapuas region, can differ quite significantly over a short distance. Due to this large variation, rain gauges will have a small representative area. As TRMM measures the total rainfall over an area of $0.25° \times 0.25°$, the measured amounts are expected to represent the average rainfall over that area. The seventeen-year record of TRMM rainfall estimates confirms the rainfall pattern over the Kapuas and the Mahakam catchments to be bimodal with the average annual rainfall of approximately 3630 mm and 3000 mm, respectively. Zoomed into the lowland areas, the Kapuas wetlands receives an average annual rainfall of 3700 mm, while the Mahakam wetlands receives an average annual rainfall of 2690 mm. The peak of rainfall usually occurs in December-January, with the second peak in March-April and the driest month around June-August. This pattern is ever-shifting backward/forward among others due to the ENSO influence.

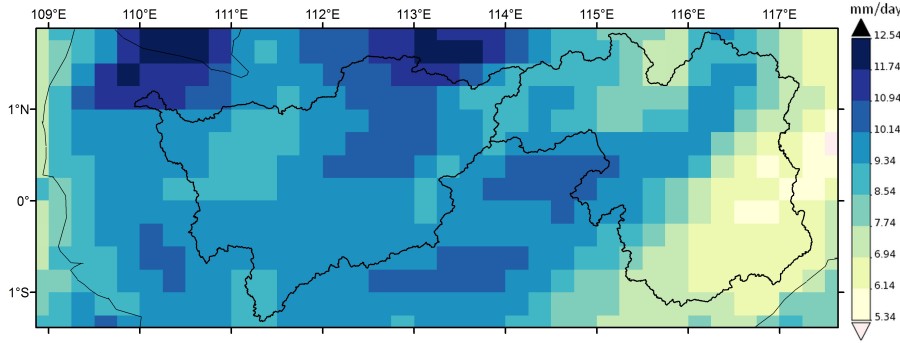

**Figure 6.** Spatial distribution of rainfall as TRMM daily rainfall rate (mm) averaged over January 1998 – December 2014. Bold black lines indicate catchment boundaries and thin lines indicate approximate coast lines.

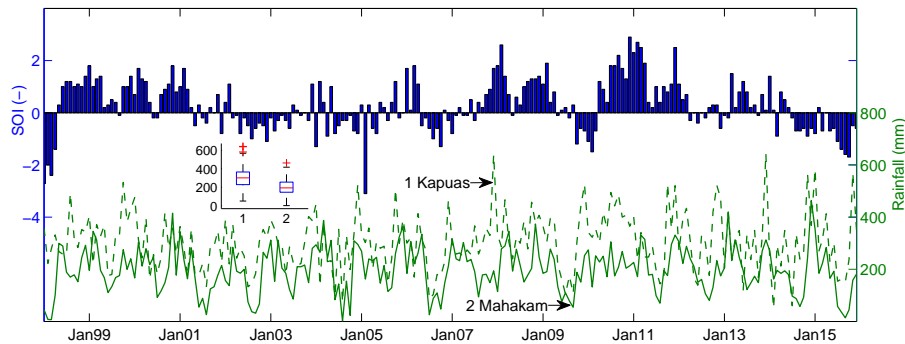

**Figure 7.** Monthly rainfall of the Kapuas (dashed line) and the Mahakam (continuous line) lowlands from January 1998 through December 2015 plotted along with the Southern Oscillation Index (bars). Box plots show the variability of monthly rainfall in the two lowland regions.

The influence of ENSO in the region results in significant annual variation of rainfall. Rainfall is generally low in Indonesia during the warm period of ENSO indicated by a negative Southern Oscillation Index (SOI) and the opposite trend occurs when the SOI is positive. Figure 7 shows the relationship between SOI and monthly rainfall depth in the Kapuas and Mahakam wetland regions (red boxes in Fig. 1) with correlation coefficients ($r$) of 0.212 and 0.358, respectively. This correlation, however,
5   is not uniform throughout the season. For the Kapuas wetlands the highest $r$ value of 0.725 is found in August, while for the Mahakam wetlands the highest $r$ value of 0.732 is found in October. These r coefficient values show that the Mahakam wetland is more affected by the ENSO, which is also confirmed by its rainfall rate and drought duration described in the respective Sub-Section. From the figure, an increasing trend in the peak of rainfall in the two catchments is also observed, while the trends of the mean and minimum rainfall are hardly detected from the available TRMM rainfall estimates.
10   Groundwater and soil moisture respond relatively quickly to rainfall events. Figure 8 (top panel) shows the increase/decrease of water level and soil moisture as results of rainfall/no rainfall events at one of our monitoring stations in the upper Kapuas area. A similar result was found in the upper Kapuas peat-dominated Bika sub-catchment as reported by Wuis (2014). This

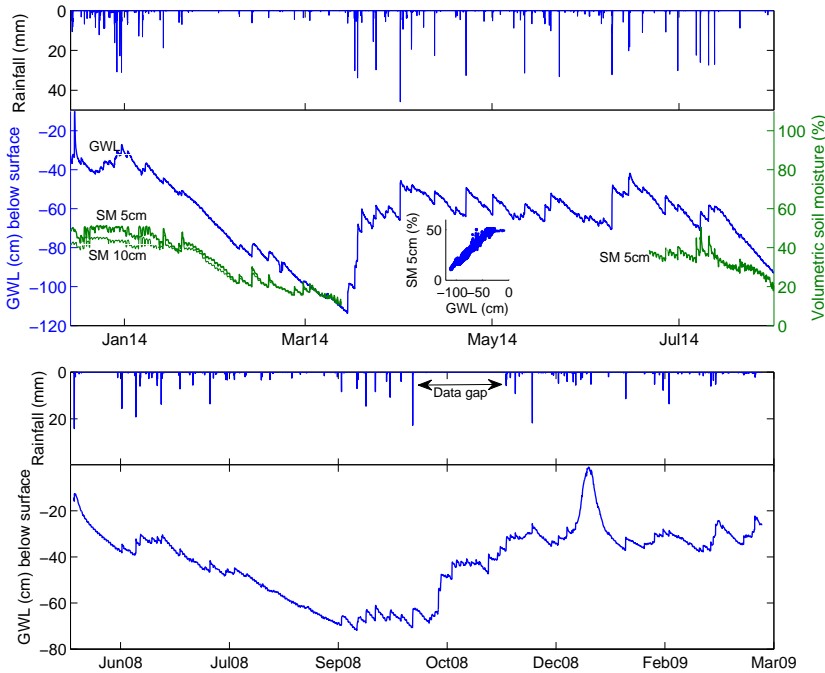

**Figure 8.** (top) Hourly groundwater level and surface (5 cm deep) and 30 cm deep soil moisture response as results of rainfall (and no rainfall) events in the Kapuas. The scatterplot shows the relation between soil moisture and groundwater level. Note that while rainfall and soil moisture were measured at the same location (mineral soil), groundwater level was measured about 5 km apart (peat soil). (bottom) Hourly rainfall and groundwater level of the Mahakam peat forest nearby Lake Melintang.

sub-catchment is a fast responding system characterized by shallow groundwater levels and rapid groundwater fluctuations; the increase in groundwater levels is clearly visible after precipitation events. Soil was saturated during the first period of our measurement in December-January, which is the peak period of rainfall (Fig.7) followed by reasonably dry conditions in February that lead to the dry-out of the lakes in the Kapuas wetlands in March 2014 (Fig. 2, bottom right). Groundwater and

5   soil moisture are well-correlated. Although the location of measurements are about 5 km apart; soil moisture was measured in a mineral soil while groundwater level was measured in a peat soil. No soil moisture observation was carried out in the Mahakam. Figure 8 (bottom panel) shows hourly rainfall and groundwater level measured at a peat forest site next to Lake Melintang. During a normal period, groundwater level response to rainfall events is rather lenient due to the presence of the lake nearby. The response is slightly faster during dry conditions as the lake's water level is also low. As the Melintang peat

10  forest is part of the Mahakam lowlands, groundwater level is not only a function of local rainfall but it is also influenced by upstream conditions. River bank overtopping upstream of the lake region causes a sudden increase in water level observed during the peak of the wet period.

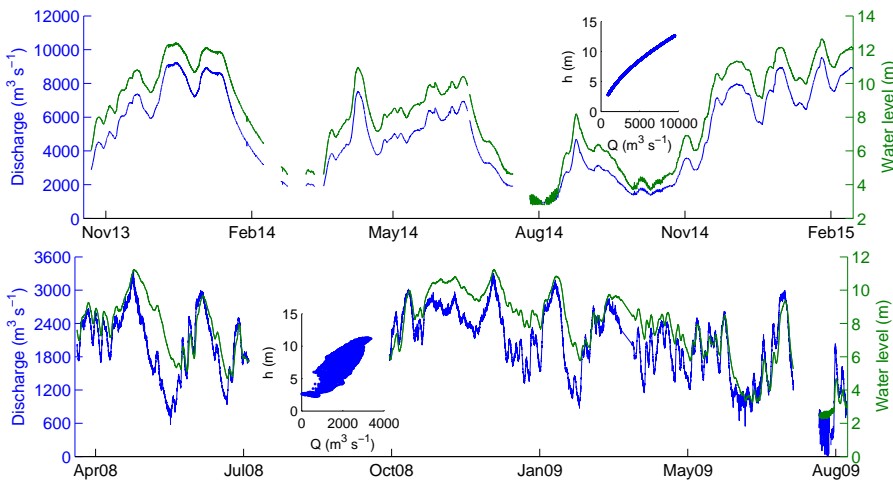

**Figure 9.** Discharge series from flow monitoring stations of the Kapuas at Sanggau (top) and the Mahakam at Melak (bottom) with scatter-plots showing stage-discharge relations. Water stages are with respect to the pressure gauges position.

## 4.2 Streamflow and flood hydraulics

Consistent with the local humid climate, the Kapuas and the Mahakam are rainfall driven rivers. Figure 9 shows discharge series of the Kapuas at Sanggau and the Mahakam at Melak flow monitoring stations. Hydrograph behaviour of the Kapuas at Sanggau and the Mahakam at Melak generally correspond to monthly average precipitation behaviour with a small delay in time. The Kapuas mean discharge at Sanggau is about 5000 $m^3s^{-1}$, while that of the Mahakam at Melak is about 2000 $m^3s^{-1}$. During low flows in July-August, both discharge time-series show sub-daily fluctuations, which correspond to tidal signals. Tidal energy can reach that far upstream because the terrain of the middle and lower regions of the Kapuas and that of the Mahakam are relatively flat.

The Kapuas flow monitoring station at Sanggau is located downstream of the wetlands area and lies in relatively steeper terrain that marks the transition between the middle and the lower Kapuas regions. This results in an almost unambiguous stage-discharge relation due to the absence of backwater effects. The opposite condition was observed in the discharge series from the Mahakam discharge station at Melak that was located upstream of the wetlands area in a relatively mild topography. Variable backwater effects from floodplain impacts, lakes and tributaries, and effects of river-tide interaction are apparent in the ambiguous stage-discharge relation. Water level fluctuations of about 10 m were recorded during our observation at Melak station. Discharge at this location is highly hysteretic, which is extensively discussed in Hidayat et al. (2011b). From water level records, they show that wetlands including some thirty lakes in the middle Mahakam area play a role in water level peak attenuation via a lake filling and emptying mechanism.

Water levels also fluctuated with a range of about 10 m at our discharge monitoring station of the Kapuas at Sanggau. No significant hysteresis occurs in the stage-discharge relation at this flow monitoring station. As it is the case of the middle Mahakam region, backwater effects are likely to occur upstream of the Kapuas wetlands area, which was investigated using

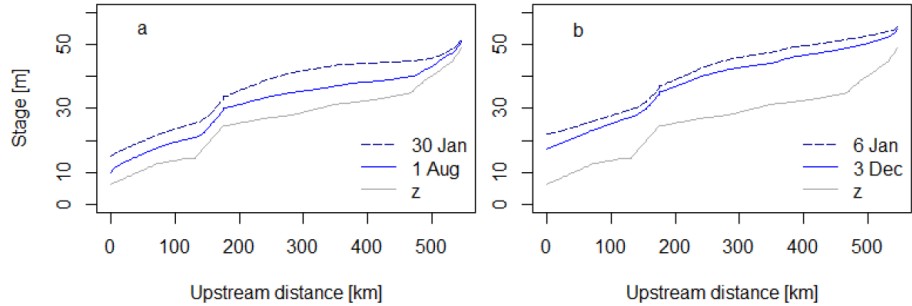

**Figure 10.** Longitudinal surface water profile over the Kapuas river reach from Sanggau to Putussibau; z is bed elevation. (a) Water stages are plotted for 30 January and 1 August displaying the falling limbs of the hydrograph. (b) The rising limbs, when the water levels are increasing, are shown for 3 December 2014 and 6 January 2015.

the HEC-RAS river analysis system (Bol, 2015). Figure 10 (left) shows the falling limb of the hydrograph from the Kapuas HEC-RAS model simulation. When the water levels drop, a smooth surface water profile develops. In the upper half of the Kapuas a water hump is formed where the maximum water depth of 13.1 m occurs about 280 km upstream of Sanggau. The rising limb of the hydrograph is displayed in Fig. 10 (right). At about 350 km upstream of Sanggau the Tawang channel meets

the Kapuas, connecting the wetland to the river. Directly upstream of this wetland a backwater curve arises during the rising limb of the hydrograph. Wuis (2014) observed hysteresis loops in the stage-discharge relation of the Bika River, a tributary debouching to the Kapuas upstream of the wetland area. She found that due to backwater effects from the main Kapuas as well as from ponding of the lower Bika region, low water levels can imply both a low and a high discharge, which also holds for high water levels.

**4.3  Inundation dynamics**

Both catchments are characterized by vast wetland areas. Figure 11 shows the inundation occurrence in the Upper Kapuas and Mahakam lakes area derived from 20 PALSAR images available in the period of 2007 through 2010. Estimates of the total volumes of the lakes in the Kapuas and the Mahakam lakes area are shown in Fig. 12. As these estimates were derived without incorporating the lake's bathymetry, which is generally flat as shown on Fig. 2 (bottom right), the value presented herein should

be considered merely as a rough estimate of storage capacity. The highest estimated total volume of water in the Kapuas lakes is about 3 billion $m^3$ during the PALSAR data acquisition on 8 April 2009, while that of the Mahakam lakes is about 6.5 billion $m^3$ during the PALSAR data acquisition on 12 May 2010. Considering the extent of flooding under vegetation cover (Fig. 11), the maximum total volume of water stored in these wetlands can be twice as much as the abovementioned values. This partly explains why such a large discharge variation occurred at a given water stage as shown in the scatterplot of Fig. 9 (bottom

panel) at the discharge station upstream of the wetland region.

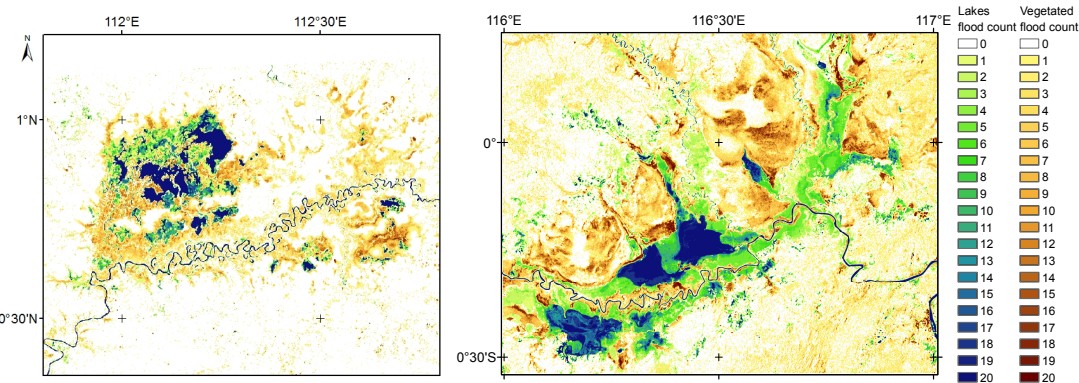

**Figure 11.** Inundation occurrence in the Upper Kapuas (left) and Mahakam (right) lakes area derived from 20 PALSAR images of 2007 – 2010 as open water (light green – dark blue) and flooding under vegetation (light – dark brown). Legend vertical at the right: "Lakes flood count" in "Open water"; "Vegetated flood count" in "Flooding under vegetation".

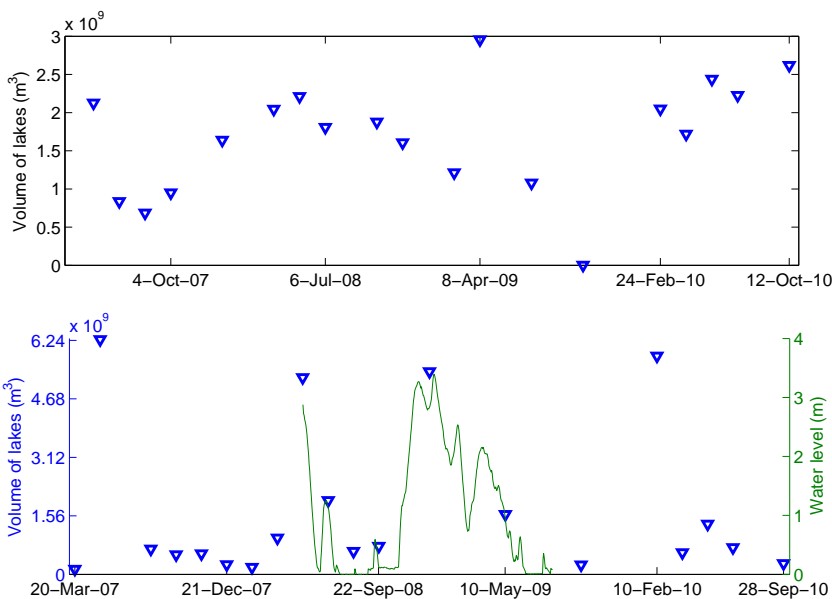

**Figure 12.** Estimates of total volume of lakes in the Kapuas (top) and the Mahakam (bottom) lowlands derived from flooded pixel-areas in PALSAR images. Note that stream network was not removed during the flood count assessment, which renders the area of inundation to be somewhat overestimated.

## 4.4  Drought occurrence and fire vulnerability

Groundwater tables in the lowlands of Borneo are typically shallow; capillary rise from groundwater normally feeds soil water in the topsoil to (partially) compensate for the high evapotranspiration losses. Groundwater recharge reflects well soil

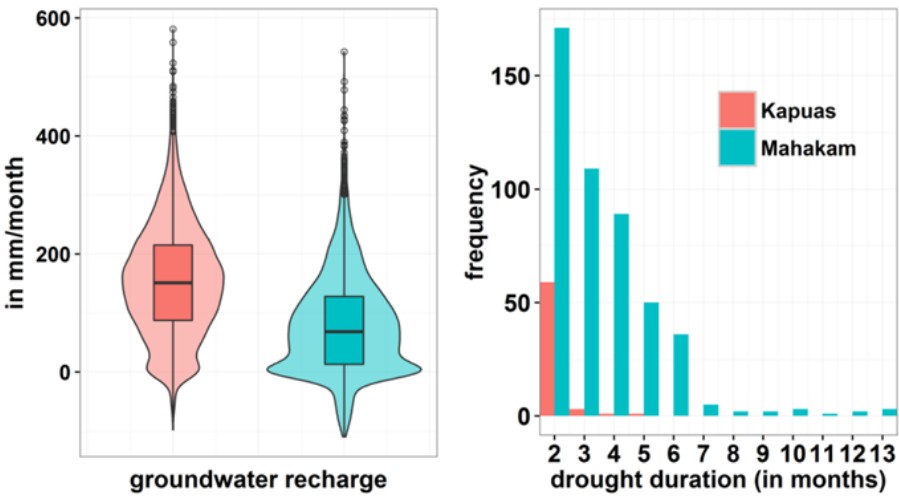

**Figure 13.** Hydro-climatological regimes and drought in the Kapuas and Mahakam lowlands: (left) Violin and box plots showing the distribution of monthly groundwater recharge over the period of the TRMM dataset (1998-2014); (right) drought in groundwater recharge showing different durations.

hydrological processes, incl. capillary rise and evapotranspiration. During the dry season, low recharge is common and it can even become negative, meaning topsoil moisture is too low to compensate for the evapotranspiration flux and capillary rise occurs, which is not always enough to keep the evapotranspiration at the potential rate. The two lowlands under study show a different hydro-climatic regime. In the Kapuas, a high median monthly groundwater recharge (161 mm/month) occurs, which is twice as high that of Mahakam lowland (Fig. 13, left panel).The Kapuas lowlands have few periods with low groundwater recharge during the dry season. Prolonged drought is rarely found in the Kapuas (only few events last up to 3 months (Fig. 13, right panel). In contrast, the Mahakam wetland area regularly exhibits low groundwater recharges, which may lead to prolonged drought events that can last up to 13 months. It appears that the Mahakam lowland is more vulnerable to hydrological drought, leading to higher risk at fire occurrence.

## 5  Discussion

Tropical rivers exhibit a fairly direct response to rainfall input with strong seasonality as a dominant feature (Syvitski et al., 2014). This is also the case in the Kapuas catchment that generally reacts very quickly and behaves as a straightforward system. In a lowland region, the presence of wetlands locally averts the river system from this general principle. Our HEC-RAS model simulations shows that lakes and adjacent wetlands have a delaying effect on river discharges and diminish water levels and flow fluctuations. The average lake level fluctuation of 7.3 m for the entire wetland area, are in the same range as previously documented ranges of 8 m and 11 m (Klepper et al., 1996; Anshari et al., 2004). During the rising limb of the hydrograph, as could be seen in Fig. 9, a backwater curve arises just upstream of the Tawang channel that connects the lakes area to the

Kapuas. Next to discharge, backwater influences river flow velocities, which is an important factor in river meander growth. From an analysis using a series of Landsat images, Huisman (2014) found a clear transition between a more active channel migration of the Kapuas upstream of the lakes area and a less active migration downstream. The border between these two regimes was found 10 km upstream of the Tawang channel. At this border, a clear transition between river flow velocities is observed. Erosion and transport of the eroded material can take place during bankfull discharge, while during low flows, when flow velocities fall below the settling velocity of sediment particles, sediments can be deposited. The distinct changes in flow velocities and the accelerating and decelerating behaviour could be an extra explanation why the Kapuas is very actively meandering in its upper reaches. During the rising limb of the hydrograph no distinct effect of the wetland can be seen at Sanggau discharge station. The peak flows occur at the same moment in time and the discharge rises with the same speed. During the falling limb, effects of the wetland can be seen. The falling limb occurs more gradually and the minimum flow changes due to the supply of water from the lake.

Next to the effect on the river discharges, the wetland also gave rise to a backwater curve, arising upstream of the Tawang channel. This backwater curve affects the flow velocities and can influence the meandering behaviour of the Kapuas river. Processes of lake filling and emptying contribute to accelerating and retarding the river flow velocity. Due to the backwater effect from the lake, when lake level is high, the stage upstream of the lakes area is relatively high for a relatively low discharge. This effect is reduced when the lake level dropped and the discharge increased while the water level keeps decreasing, pending sufficiently high discharge that renders water level to follow the discharge trend. During lake filling process, the contrasting mechanism took place. These mechanisms play a key role in regulating water level and discharge downstream (Hidayat, 2013). Vast areas of lowland wetlands in the middle Kapuas as well as in the middle Mahakam region form a massive water storage that eliminates sudden and large river discharge changes downstream. The moderation of discharge and water level fluctuations in the lower reaches of the river by the lake filling and emptying mechanisms as shown by water level records as well as river-tide interaction results in a relatively mild discharge variation in the downstream region (Hidayat et al., 2011b; Sassi and Hoitink, 2013).

The accurate, continuous discharge estimates from the H-ADCP allow for a discussion on the difference between runoff and discharge in the backwater affected region, in our case upstream of the wetland/lakes region. ADCP flow measurements are costly and time consuming. Nevertheless, there are no simple alternative to monitoring discharge dynamics in rivers with back-water. Too often, water agencies rely on rating curves that fail to capture the hysteresis effects. Hydrologists may not always be sufficiently critical about the accuracy of discharge estimates from rating curves. The data series presented in this contribution is relatively short, but we would like to point to the fact that regarding discharge data, long data series available elsewhere are always based on rating curve information, whereas our observations are made independent for discharge (using an H-ADCP) and water level. The rainfall data period is extended by the availability of TRMM products and potential evapotranspiration from CRU, here we use data from 1998 through 2015. We concur that the analysis presented can be improved in the future, when longer data series are available, but the present data series are long enough to support the conclusions drawn. Further, the influence of ENSO on the Kapuas and Mahakam is not equally significant. This knowledge would drive to develop different consequences and policies.

The present study make a systematic inventory of the existing studies of inland tropical lowlands, which have received relatively little attention, including gathering continuous flow data to accurately estimate river discharge, one of key hydrological variables. Concerning discharge of the Kapuas, we are the first to record water discharge on the Kapuas River based on H-ADCP flow monitoring. Once a representative length of flow measurements is obtained, a model can be constructed to relate discharge data with other simpler to measure parameters such as stage, known as the rating curve technique. However, rating curves are subject to uncertainties concerning interpolation and extrapolation errors and seasonal plants variations (Di Baldassarre and Montanari, 2009) and may fail to capture discharge dynamics in lowland river reaches affected by backwater effects due to the inapplicability of the kinematic wave equation to handle the surface gradient term in the non-inertial wave equation (Hidayat et al., 2011b). A novel technique, such as neural networks, can be applied to model discharge in this hydrologically complex region. Hidayat et al. (2014) demonstrate that discharge in a tide dominated river reach can be well-modelled even without at site stage records given a representative discharge time-series is available for training and validating the model. This implies that an H-ADCP can be installed temporarily in one location, e.g. one year of flow measurements, to obtain discharge estimates that can later on be used to train and validate the model. Once the model is established, it can be used for discharge prediction whilst the H-ADCP can be installed in another location, optimizing investment in monitoring instrumentation, even though a permanent H-ADCP station would be ideal. Similar to that of the traditional rating curve technique, however, occasional updating with new data is required to adjust the neural network model to account for changes in the river system.

As radar is unrestricted by cloud cover, radar remote sensing technology presents an alternative to detect changes in inundation states of wetlands in the humid tropics. From its dark signature, a fully inundated region can be easily recognized on radar images. From combining images produced by taking the minimum, mean, and maximum backscatter values of radar images, a clear signature of flooding under vegetation can be obtained and from such images floodplain delineation can be performed (Hidayat et al., 2011a). The results presented herein provide a basis for a better understanding of the role of the Kapuas and the Mahakam inland wetlands, in buffering the discharge towards the downstream coastal regions. Radar-based floodplain observations may be used in future work to calibrate hydrodynamic models simulating the filling and emptying processes of the lakes area. This work has also contributed to the understanding of tropical lowlands. We find two important features, namely 1) widespread flooding and strong surface water-groundwater linkage, and 2) strong backwater effects that form a dramatic multiple discharge hysteresis as shown in the $Q$ - $h$ plot. We reveal the impacts of backwater effects, which proves the kinematic wave approach adopted in many hydrological studies unsuitable. These findings imply that many hydrological models will fail to describe the hydrology correctly if they do not account for the presence of standing water or high groundwater tables, and that many simple routing routines will fail to describe the discharge dynamics.

Climate extremes, both wet and dry, can have devastating impacts in tropical regions. Such impacts are well-documented for the Amazon basin, but have received less attention for tropical rainforests in Asia, in particular those in Kalimantan. An exception forms the large-scale drought-induced wildfires that have occurred over the past decades, and which are associated with the El Niño Southern Oscillation (ENSO) phase (Siegert et al., 2001). Before the 1980's, it was believed that tropical rainforests were resilient to drought. During the 1982/1983 El Niño, it became clear that prolonged drought could cause

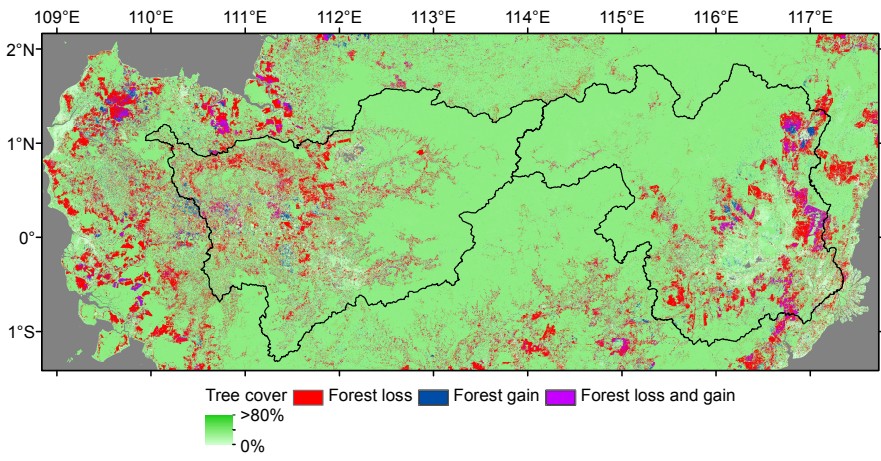

**Figure 14.** Tree cover (green) forest loss (red) forest gain (blue) with loss and gain (purple) between 2000 – 2012 in the Kapuas and the Mahakam. Data source: Global Forest Change (Hansen et al., 2013).

large-scale damage to rainforests and peat soils due to widespread wildfires (Malingreau et al., 1985). Wildfires spread more easily due to accumulation of leaf litter (van der Molen et al., 2011), which become extremely dry during drought conditions. This factor, along with those of anthropogenic origin e.g. the conversion of forests to other land uses, is the main contributor to loss of rainforest in Kalimantan's lowlands (Fig. 14). 72-85% of the middle Mahakam peatlands were burned between 1997 and 2000 (Chokkalingam et al., 2005). Fires mostly occurred within this period during the 1997/1998 El Niño-induced extreme dry condition. Another study by Siegert et al. (2001) reported that more than 2 million Ha of forest were burned in lowland Mahakam. At present, large-scale wildfires occur almost every year during the dry season, raising the questions whether climate variability and circulation changes can amplify anthropogenic land use changes, and how this will impact the hydrological functioning of the study area.

## 6 Conclusions

Alluvial floodplains of lowland rivers have become the centre of past and present human settlement due to their fertile soil that supports food production and easy access for transportation. With the ever-increasing water demands and threat from water-related disasters, hydrological prediction, on the one hand, is crucial to support a resilient society inhabiting the area. On the other hand, hydrological predictions in lowland rivers especially in tropical regions are difficult considering the scarcity of hydro-meteorological data and flow regime complexity resulting from, among others, lake-river interaction, backwater and tidal effects. This study offers a comprehensive view on the hydrological characteristics of two poorly-gauged tropical inland lowland rivers: the Kapuas and the Mahakam in Kalimantan, Indonesia. Based on TRMM data, it was shown that both river basins experience strong seasonal fluctuations in precipitation. The Kapuas basin receives considerably more precipitation than the Mahakam. The Kapuas wetland area receives an average annual rainfall of 3700 mm, while the Mahakam wetland

area receives an average annual rainfall of 2690 mm. In response to the strong seasonal variations in water input, both basins showed a strong seasonal variability in inundation extent as derived from PALSAR images. The Kapuas and Mahakam lakes regions are vast reservoirs of water that can store as much as 3 billion $m^3$ and 6.5 billion $m^3$ of water, respectively, which can be doubled when the area of flooding under vegetation cover is considered. We found the seasonally varying storage in both wetlands to exhibit an important role in regulating the discharge regime of the downstream parts of the rivers. Directly downstream of the wetland, the river discharges are most clearly affected. Based on discharge observations made by H-ADCP during dedicated field campaigns over multiple seasons, we found strong dynamics in both discharge and water levels. The seasonal amplitude in water levels was found to be around 10 m for both basins. Strong backwater effects in the Mahakam prohibited the use of a traditonal rating curve for discharge estimation, calling into question the quality of historical discharge records in many lowland basins. Contrary to the moist nature of wetlands, the two lowlands are vulnerable to drought especially during the warm period of ENSO, yet prolonged drought rarely occurs in the Kapuas under current climate condition in line with observations of shallow groundwater tables and a strong coupling between groundwater and soil moisture observations. We found that the Mahakam lowland area is more vulnerable to hydrological drought leading to fire occurrence. It is expected that the hydrological characterization of the Kapuas and the Mahakam facilitates better prediction of fire-prone conditions in these regions.

Our observations and analysis reveal a region dominated by highly dynamic hydrological processes, such as seasonal inundation over vast areas ($> 1000$ km$^2$), strong backwater effects, shallow groundwater tables and a high seasonal amplitude of river stages ($\sim 10$ m). Most of these processes are currently not or only crudely represented in hydrological models. We believe our study can contribute to the use of data from poorly-gauged catchments to improve the next generation of models in areas that were traditionally a "blind-spot" for model evaluation, but where strong changes in land use and climate provide an urgent need for better models.

*Acknowledgements.*  This research has been funded by The Royal Netherlands Academy of Arts and Sciences (KNAW) through the Scientific Programme Indonesia-Netherlands SPIN3-JRP-29 and by the Netherlands Organization for Scientific Research (NWO grant number WT76-268). ALOS PALSAR data have been provided by JAXA EORC within the framework of the ALOS Kyoto and Carbon Initiative. TRMM data analysis and visualization used in this paper were produced with the Giovanni online data system, developed and maintained by the NASA GES DISC.

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
