# Peer review of "Hydrology of inland tropical lowlands: The Kapuas and Mahakam wetlands"

_Hydrology and Earth System Sciences, 2016_

## Referee Comment (RC1) · K. Hassaballah (Referee) · 28 Sep 2016

General comments:

This manuscript is about the analysis of hydrological dynamics of two neighbouring poorly gauged tropical basins; the Kapuas basin (98,700 km2) in West Kalimantan and the Mahakam basin (77,100 km2) in East Kalimantan, Indonesia. Although the topic is interesting and relevant for this journal, the knowledge gap that the manuscript aims to bridge is not clear. The authors gave only general description of the study area without paying attention to show the value of the two wetlands for local communities as well as for the country, which is very important for the significance of the study. Also maybe authors could describe the problem a little bit more, so that the readers can easily understand the gravity of the situation. It is clear that the authors have made a

substantial effort for collecting field data. However, the methodology has described the collected data, but does not clearly describe the purpose for which the collected hydro-climatological field data were used. Another issue is that, calibration and validation are essential for hydrological modelling before simulating any hydrological processes. Please add a section describing how do you calibrate and validate your HEC-RAS model. Many other specific and technical comments can be found below.

Specific comments: (P=Page, L=Line)

1. P1, L6-7 and P6, L2: The authors have mentioned that the hydro-climatological data were obtained during fieldwork campaigns carried out in the Mahakam over the period 2008-2010, while table1 shows that all data in this catchment was collected between Feb2008-Aug2009.

2. Is this short period of measurements (2008-2009 in Mahakam and 2013-2015 in Kapuas) sufficient enough to capture the hydrological variability in the study area?

3. The hydro-climatological data were carried out separately in the two catchments (in different hydrological years), during 2013-2015 in the Kapuas basin, while in the Mahakam the data were collected during 2008-2009. Is this has any effect on the overall results when comparing results from the two catchments (i.e. comparing vulnerability to hydrological drought)?

4. P1, L8-12: In this paragraph, it is clearly that rainfall estimates from the Tropical Rainfall Measuring Mission (TRMM) was used for analysing the distribution of rainfall and the influence of El-Niño – Southern Oscillation. Flood occurrence maps were derived from the PALSAR images. But it is not clear for which purpose the collected hydro-climatological field data (e.g. rainfall, air temperature, relative humidity, solar radiation, wind speed and direction) was used. Is it for model calibration and validation?

5. P5, L7-9: The Authors reported that due to the global circulation and the regional climate the central and northern parts of Kalimantan have bimodal rainfall patterns with

two peaks of rainfall. Those peaks generally occur in October through November and March through April. Is this consistent with the results of the HEC-RAS model shown in Figure 8 which presented the timing of rising and falling limbs?

6. P6, L6-7: Why both daily 3B42 and monthly 3B43 TRMM rainfall products were used?

7. P7, L8: Do the Authors have any explanation for why a spatial discretization of 3 km and a time step of 20 minutes were chosen to run the HEC-RAS model?

8. P7, L10: What is the accuracy of PALSAR images for estimating the inundated area?

9. P7, L26-27: What is the resolution of the potential evapotranspiration?

10. P8, L 15-16: "Figure 5 shows the relationship between SOI and monthly rainfall depth in the Kapuas and Mahakam wetland regions with correlation coefficients (r) of 0.212 and 0.358, respectively". Please discuss the physical meaning of such correlations?

11. P17, L11-12: The authors state in their conclusion that this study highlights the merits of H-ADCP continuous flow measurements to obtain accurate discharge estimates when rating curves fail. Using ADCP for flow measurement is not new, but costly and time consuming.

12. How frequent are the discharge measurements using the HADCP, and why?

Technical corrections:

1. P3, L33: Where is the location of Putussibau on the map?

2. P5, Figure 2 caption's: better indicate the exact year (e.g. during 2013) instead of "during the previous wet period".

3. P6, L12: For "Horizontal Acoustic Doppler Current Profiler" Please use uppercase

for all abbreviated words.

4. P6, Table 1: groundwater instead of (ground)water. The same on P10, L1.

5. P6, L15: Either remove the dash from (H-ADCP) to be consistent with the previous abbreviation, or add a dash to (HADCP) when first mentioned.

6. P8, L14: Spell out the first use of SOI in the main body of the manuscript.

7. P9, L1-2: Better use back slash "/" between two opposite processes (e.g. increase/decrease) instead of increase (decrease). The same for rainfall/no rainfall instead of rainfall (no rainfall).

8. P9, L6: The peak monthly rainfall has shown in Figure 5 instead of Figure 4.

9. P9, L7-8: "Groundwater and soil moisture are well-correlated. Although...etc" instead of "Groundwater and soil moisture are well-correlated although...etc".

10. P9, L10: Figure 6 instead of Figure 5 (hourly rainfall and groundwater levels were presented in Figure 6). The authors should pay more attention when referring to figures.

---

## Referee Comment (RC2) · M.C. Westhoff (Referee) · 1 Nov 2016

In this paper, the authors look at the hydrology of two poorly gauged wetlands. They combined point data of soil moisture and groundwater levels and discharge data from a downstream station with remote sensed data of rainfall and images for deriving the lake area.

However, I missed a clear objective or research question. The title says that this manuscript deals with the hydrology of these wetlands, but no clear explanation is given which part of the hydrology is considered. In fact, the Results section is absent and replaced by the 'Hydrological characterization' section.

To me this manuscript reads like a synthesis of previous findings with just some additional measurements. Most 'novel' methods or insights have (apparently) already been

published previously: e.g. the use of PALSAR images to estimate flood area has been discussed in Hidayat et al. (2011, 2012); the influence of backwater at the discharge location has been discussed in Hidayat et al. (2010) and Wuis (2014); The filling and spilling of the lake has been discussed in Hidayat (2013); the construction of a rating curve with a neural network model and only a short period of data has been presented by Hidayat et al. (2014).

I was also rather surprised when I read the in the conclusions that 'The present study highlights the merits of H-ADCP continuous flow measurements to obtain accurate discharge estimates when rating curves fail'. To me, this statement came out of nowhere: In the manuscript two discharge curves and rating curves have been shown, but no accuracy of this method has been reported, nor has it been compared with other methods. Instead the authors mainly suggest that it is possible to move the device after a year of observations after which the neural network model of Hidayat et al. (2014) can be applied to construct a rating curve.

Also, as one of their main findings, the authors state that: "This work has also contributed to the understanding of tropical lowlands. We find two important features, namely 1) widespread flooding and strong surface water-groundwater linkage, and 2) strong backwater effects." However, these features are characteristic for all kind of wetlands – not only tropical ones. In the same manuscript this is already mentioned in the introduction (P2, L10-14).

Altogether, this lack of novel aspects and the fact that no clear objective is given is reason for me to advice rejection of the manuscript.

References:

Hidayat: Runoff, discharge and flood occurrence in a poorly tropical basin: The Mahakam River, Kalimantan, Ph.D. thesis, Wageningen University, 2013.

Hidayat, Hoekman, D. H., Vissers, M. A. M., and Hoitink, A. J. F.: Combining ALOS-

PALSAR imagery with field water level measurements for flood mapping of a tropical floodplain, Proc. SPIE 8286, Eds.: X. He, J. Xu, V. G. Ferreira, International Symposium on LIDAR and Radar Mapping: Technologies and Applications, Nanjing, China, doi:10.1117/12.912735, 2011.

Hidayat, H., Hoekman, D. H., Vissers, M. A. M., and Hoitink, A. J. F.: Flood occurrence mapping of the middle ahakam lowland area using satellite radar, Hydrol. Earth Syst. Sci., 16, 1805–1816, doi:10.5194/hess-16-1805-2012, 2012.

Hidayat, H., Hoitink, A. J. F., Sassi, M. G., and Torfs, P. J. J. F.: Prediction of Discharge in a Tidal River Using Artificial Neural Networks, J. Hydrol. Eng., 19, 04014 006, doi:10.1061/(ASCE)HE.1943-5584.0000970, 2014.

Wuis, M.: Hydro(geo)logical flow characteristics of the Bika catchment, a subcatchment of the Kapuas River, Indonesia: A reconnaissance study, Master's thesis, Wageningen University, 2014.

———————————————

---

## Author Comment (AC1) · 25 Nov 2016

The Authors highly appreciate suggestions and constructive criticisms posed by the Reviewers. We also would like to thank Prof. Stefan Uhlenbrook for handling the review process of the manuscript. Here we present our response to the discussion issues that have arisen during the review process.

1. Reviewer #1

We thank Khalid Hassaballah for his constructive comments and suggestions. We are pleased to see that most remarks concern the way we have written the Introduction, and few remarks concern the methodology and/or the robustness of the results. Below we include point-by-point replies to the reviewer's comments.

[Figure]

General comments:

Comment: Although the topic is interesting and relevant for this journal, the knowledge gap that the manuscript aims to bridge is not clear. The authors gave only general description of the study area without paying attention to show the value of the two wetlands for local communities as well as for the country, which is very important for the significance of the study. Also maybe authors could describe the problem a little bit more, so that the readers can easily understand the gravity of the situation.

Reply: In retrospect, the Introduction of our Discussion paper indeed lacks the main motivation for our work, and can be revised to become more explicit in identifying the relevant knowledge gaps. The motivation or relevance itself was not questioned. In a revised version, we will add more text explaining the importance of the two wetlands in the Introduction. Lake Sentarum National Park in the upper Kapuas River is an important Ramsar site, which represents one of old tropical peat formations in Late Pleistocene (Anshari et al 2001, 2004). The Mahakam River is home to endemic species including the Irrawady dolphin (Orcaella brevirostris), which is listed as 'Vulnerable' on the International Union for Conservation of Nature (IUCN) Red List of Threatened Species due to, among others, entanglement in gillnets, vessel traffic, sedimentation, habitat loss and degradation from habitat change (IUCN, 2016). The Kapuas and the Mahakam wetlands are important for their respective local communities not only as a source of water for domestic purposes, but also to sustain the livelihood of the people especially in the open water fishery sub-sector. The Kapuas wetland with its seasonally inundated lakes produces about 18,000 tons of freshwater fish annually (BPS-Kalbar, 2015). The Middle Mahakam wetland is the core of inland fisheries in East Kalimantan and is considered as one of the most productive freshwater fisheries in Southeast Asia (Mackinnon, 1996) with a current estimated annual fish production of 33,000 tons (BPS-Kaltim, 2015). These fishing industry figures express the high economic value of the wetland areas we study. The unique tropical wetland ecosystems are rich in biodiversity of typical aquatic as well as terrestrial flora and fauna, which is why they

are listed as a Ramsar site, urging conservation. The wetlands act as hydrological buffers and hence contribute to natural flow regulation, which has rarely been quantified. There is a lack of knowledge on processes involved in seasonal flooding in tropical lowland areas, including backwater effects. Notwithstanding their ecological, hydrological and economical importance, the Kapuas and Mahakam wetlands in particular and the two river basins in general have been increasingly threatened by a variety of factors including pollution, forest fires, deforestation, and mono cultures (Rautner et al., 2005; Chokkalingam, 2005).

Comment: The methodology has described the collected data, but does not clearly describe the purpose for which the collected hydroclimatological field data were used. Another issue is that, calibration and validation are essential for hydrological modelling before simulating any hydrological processes. Please add a section describing how do you calibrate and validate your HEC-RAS model.

Reply: The first part of the comment is similar to the observation of Reviewer #2, who argues that the knowledge gaps and purpose of the campaign was not yet sufficiently motivated. The Kapuas is the largest river in Indonesia, which has remained poorly studied to date. The catchment can be considered relatively pristine, offering a view on a natural hydrological regime which serves both scientific and engineering purposes. This paper can be seen as an introduction paper of a large number of studies that will follow, conducted by two PhD candidates and two Postdocs . We find it inappropriate to include a long list of objectives set in the overarching project, but we will add more context and better describe the purpose of the collected data in the revised manuscript. The HEC-RAS model is directly used to quantify the effects of inland lakes on river discharge dynamics. We will add more explanations about the calibration and validation of our HEC-RAS model in the revised manuscript.

Specific comments: (P=Page, L=Line)

1. P1, L6-7 and P6, L2: The authors have mentioned that the hydro-climatological data

were obtained during fieldwork campaigns carried out in the Mahakam over the period 2008-2010, while table1 shows that all data in this catchment was collected between Feb2008-Aug2009.

Reply: The Mahakam fieldwork campaign indeed took place over the period 2008-2009. We have corrected this on the text.

2. Is this short period of measurements (2008-2009 in Mahakam and 2013-2015 in Kapuas) sufficient enough to capture the hydrological variability in the study area?

Reply: The data series is relatively short, but we would like to point to the fact that regarding discharge data, long data series available elsewhere are always based on rating curve information, whereas our observations are made independent for discharge (using an H-ADCP) and water level. The rainfall data period is extended by the availability of TRMM products and potential evapotranspiration from the Climate Research Unit (CRU), here we use data from 1998 through 2015. We agree the analysis we present can be improved in the future, when longer data series are available, but the present data series are long enough to support the conclusions drawn.

3. The hydro-climatological data were carried out separately in the two catchments (in different hydrological years), during 2013-2015 in the Kapuas basin, while in the Mahakam the data were collected during 2008-2009. Is this has any effect on the overall results when comparing results from the two catchments (i.e. comparing vulnerability to hydrological drought)?

Reply: For a drought study in the two basins, we simulated the transient water balance using rainfall estimates from TRMM and CRU' potential evapotranspiration for the period from 1998 through 2014 as input data, to derive groundwater recharge. Our measured data were used to validate the TRMM product. Therefore, the difference in the period of field campaigns in the Kapuas and Mahakam basins has little effect on the overall results.

[Figure]

4. P1, L8-12: In this paragraph, it is clearly that rainfall estimates from the Tropical Rainfall Measuring Mission (TRMM) was used for analysing the distribution of rainfall and the influence of El-Niño–Southern Oscillation. Flood occurrence maps were derived from the PALSAR images. But it is not clear for which purpose the collected hydro-climatological field data (e.g. rainfall, air temperature, relative humidity, solar radiation, wind speed and direction) was used. Is it for model calibration and validation?

Reply: We have mentioned all collected data from our Automatic Weather Station in Table 1, while actually not all are actually used in the analysis. We will remove unnecessary information such as solar radiation, wind speed and direction from the text.

5. P5, L7-9: The Authors reported that due to the global circulation and the regional climate the central and northern parts of Kalimantan have bimodal rainfall patterns with two peaks of rainfall. Those peaks generally occur in October through November and March through April. Is this consistent with the results of the HEC-RAS model shown in Figure 8 which presented the timing of rising and falling limbs?

Reply: Yes, our HEC-RAS simulation reflects this bimodality of the rainfall pattern in the study area, as shown in the simulated discharge (Fig. 1).

6. P6, L6-7: Why both daily 3B42 and monthly 3B43 TRMM rainfall products were used?

Reply: The daily TRMM rainfall rate was used to show the spatial rainfall variability in the study area and to be compared with previous studies cited in the manuscript (Wuis, 2014; Tekelenburg, 2014) with short period of observation (about two months). The monthly TRMM rainfall rate was used to be presented along with the publicly available SOI index, which has monthly values.

7. P7, L8: Do the Authors have any explanation for why a spatial discretization of 3 km and a time step of 20 minutes were chosen to run the HEC-RAS model?

Reply: Spatial resolution of 3 km is considered sufficient for the modelled river section of 553 km from Putussibau to Sanggau. Regarding time resolution, flow velocity and cross-sectional distance show that the resolution in time cannot exceed 25 minutes. Models runs showed that the model is stable for $\delta t$<20 min. For $\delta t$ equal to 30 minutes, the model was conditionally stable, and it is unstable for $\delta t$>1 hr. The difference between model outcomes on a 20 minute resolution vs a 1 minute resolution was negligible.

8. P7, L10: What is the accuracy of PALSAR images for estimating the inundated area?

Reply: The procedure for validation of open water inundation area and inundation under vegetation maps from PALSAR images was reported in Hidayat (2012). Open water inundation occurrence was validated using bathymetry data resulting in an accuracy of about 85%. The inundation occurrence under vegetation estimates was validated using water level measurement; the reported accuracy is 78%.

9. P7, L26-27: What is the resolution of the potential evapotranspiration?

Reply: The spatial resolution of the gridded potential evapotranspiration obtained from the Climate Research Unit is 0.5 degree.

10. P8, L 15-16: "Figure 5 shows the relationship between SOI and monthly rainfall depth in the Kapuas and Mahakam wetland regions with correlation coefficients (r) of 0.212 and 0.358, respectively". Please discuss the physical meaning of such correlations?

Reply: These r coefficient values show that the Mahakam wetland is more affected by the El-Ninõ and Southern Oscillation (ENSO), which is also confirmed by its rainfall rate and drought duration. We also mentioned in the text that the influence of ENSO is not uniform throughout the seasons, and that the strongest influence occurs during the driest period. We will elaborate more on these correlation values in the revised manuscript.

11. P17, L11-12: The authors state in their conclusion that this study highlights the merits of H-ADCP continuous flow measurements to obtain accurate discharge estimates when rating curves fail. Using ADCP for flow measurement is not new, but costly and time consuming.

Reply: The accurate, continuous discharge estimates from the H-ADCP allow for a discussion on the difference between runoff and discharge in the backwater affected region, in our case upstream of the wetland/lakes region. ADCP flow measurement is indeed costly and time consuming. Nevertheless, there are no simple solutions to monitoring discharge dynamics in rivers with backwater. Too often, water agencies rely on rating curves that fail to capture the hysteresis effects. Hydrologists may not always be sufficiently critical about the accuracy of discharge estimates from rating curves, which motivates our statement.

12. How frequent are the discharge measurements using the HADCP, and why?

Reply: The discharge data from HADCP measurements presented in the manuscript are in half-hourly time step. This was done to enable us to see the hysteretic behaviour of discharge as a result of backwater effects, which will be clarified in a revised version.

Technical corrections: We corrected the text in the revised manuscript as suggested. The corrections have helped improving the consistency and readability of the ms.

2. Reviewer #2

We thank Dr. M.C. Westhoff for his review comments. To summarize, his two main objections about the manuscript are vis-à-vis no clear objective presented in the Introduction and a lack of novel aspects, which we either disagree with or are able to accommodate by textual changes. To the Reviewer, this manuscript reads like a synthesis of previous findings with just some additional measurements.

Regarding his first objection, we guess that the lack of a clear objective in the Introduction might have influenced his further assessment of the work. Most comments deal

with the introduction being inadequate, but to the editor we like to stress that this in itself does not influence the relevance or soundness of the work. We argue that the rest of the comments should be seen in the light of a poor introduction and we state hereby that we are willing to rework most of this section, also since the comments of reviewer #1 agree on this. This paper is a joint effort serving as an introduction for two PhD projects and two Postdoc projects focussed on the Kapuas river system. We are sure that reshaping the manuscript, including the main objective to reveal land surface hydrological processes that are known to be uncertain in tropical regions due to the lack of historical observations, will do justice to the novelty of the results. Establishing hydrological processes is essential to understand the impact of changes in terrestrial hydrological and biogeochemical cycles including land degradation on water level dynamics, water quality, and ecology of these important yet vulnerable wetland regions. The interactions between wetlands and the river have implications for geomorphology, governing sediment retention and modulating peak discharges, and for estuarine processes, controlling salinity intrusion during low flow.

Regarding the Reviewer's second objection on "lack of novelty" we guess that this results from the fact we have been insufficiently explicit about the knowledge gap we address, i.e. the hydrology of inland tropical lowlands. In the revised manuscript, we will make more systematic inventory of the existing studies of inland tropical lowlands, which have received relatively little attention. The referee seems not to be aware of the many complexities involved in performing measurements of key hydrological variables over large tropical catchments with little infrastructure present. We quantify flooding both in terms of water level and volume, and we reveal the impacts of backwater effects, which proves the kinematic wave approach adopted in many hydrological studies wrong. We will be more explicit about this in the revised manuscript. For instance, we are the first to record water discharge on the Kapuas River based on ADCP flow monitoring. Further, the influence of ENSO on the Kapuas and Mahakam is not equally significant. This knowledge would drive to develop different consequences and policies.

As a final remark, the Reviewer's phrasing "some additional measurements" does not do justice to efforts of making observations in a poorly accessible region surrounding the largest river of Indonesia. The dataset will offer a solid database which will find its use in future research and engineering. Finally, the Reviewer is incorrect in his reference to the first Author's previous work, since the publications mentioned focussed only on the Mahakam river system, and not the Kapuas.

Reference

Anshari, G. Z., Kershaw, A. P., van der Kaars, S.: A Late Pleistocene and Holocene pollen and charcoal record from peat swamp forest, Lake Sentarum Wildlife Reserve, West Kalimantan, Indonesia, Palaeogeography, Palaeoclimatology, Palaeoecology, 171(3–4), 213–228, doi:10.1016/S0031-0182(01)00246-2, 2001.

Anshari, G. Z., Kershaw, A. P., van der Kaars, S., and Jacobsen, G.: Environmental change and peatland forest dynamics in the lake sentarum area, West Kalimantan, Indonesia, J. Quaternary Sci., 19, 637–655, doi:10.1002/jqs.879, 2004.

BPS-Kalbar: Kalimantan Barat In Figures, BPS-Statistics of Kalimantan Barat, http://kalbar.bps.go.id/ website/pdf_publikasi/Kalimantan-Barat-Dalam-Angka-2015.pdf, 2015.

BPS-Kaltim: Kalimantan Timur in Figures, BPS-Statistics of Kalimantan Timur, http://kaltim.bps.go.id/webbeta/website/pdf_publikasi/Kalimantan-Timur-Dalam-Angka-Tahun-2015.pdf, 2015.

Chokkalingam, U., Kurniawan, I., and Ruchiat, Y.: Fire, livelihoods, and environmental change in the Middle Mahakam peatlands, East Kalimantan, Ecology Society, 10(1), 1–17, 2005.

Hidayat, H., Hoekman, D. H., Vissers, M. A. M., and Hoitink, A. J. F.: Flood occurrence mapping of the middle Mahakam lowland area using satellite radar, Hydrol. Earth Syst. Sci., 16, 1805–1816, doi:10.5194/hess-16-1805-2012, 2012.

[Figure]

IUCN: Species of the Day: Irrawaddy Dolphin http://support.iucnredlist.org/sites/default/files/species_pdf/orcaella-brevirostris.pdf, accessed 22 Nov 2016.

MacKinnon, K., Hatta, G., Halim, H., and Mangalik, A.: The ecology of Indonesia series - The ecology of Kalimantan., Oxford University Press, 1996.

Rautner, M., Hardiono, M., and Alfred, R. J.: Borneo: Treasure Island at Risk, WWF Germany, Frankfurt am Main, assets.panda.org/downloads/treasureislandatrisk.pdf, 2005.
* * *
[Figure]

**Fig. 1.** Fig. 1. HEC-RAS simulated discharges for different cross sections in the downstream Kapuas River (left) and the upstream Kapuas River (right) of the modelled river section.